# Pain Standards for Accredited Healthcare Organizations (ACDON Project): A Mixed Methods Study

**DOI:** 10.3390/jpm11020102

**Published:** 2021-02-05

**Authors:** Concepción Pérez, Jimmy Martin-Delgado, Mercedes Vinuesa, Pedro J. Ibor, Mercedes Guilabert, José Gomez, Carmen Beato, Juana Sánchez-Jiménez, Ignacio Velázquez, Claudio Calvo-Espinos, María L. Cánovas, José A. Yáñez, Mireia Rodríguez, José L. Baquero, Elisa Gallach, Emma Folch, Albert Tuca, Manel Santiña, José J. Mira

**Affiliations:** 1Pain Unit, Quality Department, La Princesa University Hospital, 28006 Madrid, Spain; concha.phte@gmail.com (C.P.); mvinuesa@salud.madrid.org (M.V.); 2Atenea Research Group, Foundation for the Promotion of Health and Biomedical Research, 03550 Sant Joan d’Alacant, Spain; jimmy.martind@umh.es (J.M.-D.); jose.mira@umh.es (J.J.M.); 3Pain Research Group, Spanish Society of Primary Care Physicians, 28009 Valencia, Spain; iborvidal@gmail.com; 4Health Psychology Department, Miguel Hernandez University, 03202 Elche, Spain; 5Oncology Department, La Fe Hospital, 46026 Valencia, Spain; jgcodina@outlook.es; 6Oncology Department, Virgen del Rocio University Hospital, 41013 Sevilla, Spain; cbeato@hotmail.com; 7Family Medicine Department, Daroca Health Center, 28017 Madrid, Spain; jsanjin@semg.es; 8Pain Unit, Guadix High Resolution Hospital, 18500 Guadix, Spain; ignavel50@hotmail.com; 9Palliative Care Department, Rioja Salud Foundation, 26006 La Rioja, Spain; clacales75@yahoo.es; 10Anesthesiology Department, Ourense Hospital Complex, 32005 Ourense, Spain; maria.de.la.luz.canovas.martinez@sergas.es; 11Nursery Department, Carlos Haya Hospital, 29010 Malaga, Spain; jays56@yahoo.es; 12Nursery Department, Terrassa Health Consortium, 08227 Terrasa, Spain; manel45545@gmail.com; 13Scientific Coordination, Spanish Forum of Patients, 28018 Madrid, Spain; jlbaquero@forodepacientes.org; 14Psychology Department, La Fe Hospital, 46026 Valencia, Spain; gallach.eli@gmail.com; 15Nursery Department, Francolí Sociosanitary Hospital, 43005 Tarragona, Spain; efolch.gipss@gencat.cat; 16Oncology Department, Barcelona Clinic Hospital, 08036 Barcelona, Spain; atuca@clinic.cat; 17Spanish Society for Quality of Care, 33003 Oviedo, Spain; msantina@clinic.cat; 18Alicante-Sant Joan Health District, 03013 Alicante, Spain

**Keywords:** pain, cancer, quality of care, accreditation, patient-centered care, patient-safety

## Abstract

Up to 50% of cancer patients and up to 90% of those in terminal stages experience pain associated with disease progression, poor quality of life, and social impact on caregivers. This study aimed to establish standards for the accreditation of oncological pain management in healthcare organizations. A mixed methods approach was used. First, a pragmatic literature review was conducted. Second, consensus between professionals and patients was reached using the Nominal Group and Delphi technique in a step that involved anesthesiologists, oncologists, family physicians, nurses, psychologists, patient representatives, and caregivers. Third, eight hospitals participated in a pilot assessment of the level of fulfillment of each standard. A total of 37 standards were extracted. The Nominal Group produced additional standards, of which 60 were included in Questionnaire 0 that was used in the Delphi Technique. Two Delphi voting rounds were performed to reach a high level of consensus, and involved 64 and 62 participants with response rates of 90% and 87%, respectively. Finally, 39 standards for the management of cancer pain were agreed upon. In the self-evaluation, the average range of compliance was between 56.4% and 100%. The consensus standards of the ACDON Project might improve the monitoring of cancer pain management. These standards satisfied the demands of professionals and patients and could be used for the accreditation of approaches in cancer pain management.

## 1. Introduction

Pain is experienced by more than 50% of cancer patients [1,2] and by up to 90% of patients who are in terminal stages [3]. The onset of chronic related cancer pain is associated with disease progression, stage of disease, and, most importantly, an impact on the patient’s quality of life [4,5]. In addition, the underdiagnosis and undertreatment of pain in 4 out of 10 patients, due to poor documentation practices in patient records or lack of communication between the physician and the patient [6,7]. The use and monitoring of existing treatment guidelines and the utilization of the World Health Organization analgesic ladder [8] to produce adequate pain relief, and this affects the outcomes in the management of cancer pain [9]. Approximately 50% of patients with some type of cancer pain remain underdiagnosed or mismanaged, despite the many initiatives undertaken for adequate management [10].

In Spain, in 2018, there were 270,363 new cases of cancer, which resulted in 110,753 deaths [11]. Of special concern in the same year was the fact that there were 772,853 long-term survivors (survival of more than five years), which is attributable to the fact that the relative five-year survival rate of patients in all cancers is more than 60% [12]. This group of patients who are cured of the underlying oncological pathology may experience chronic residual pain [11].

Accreditations use performance data to identify, develop, and implement quality improvement initiatives that can lead to improved care and better patient outcomes. There is a long tradition of accreditation in oncology. For example, in the United States of America, the American Society Clinical Oncology (ASCO) conducts several types of accreditation, one of which, the Quality Oncology Practice Initiative (QOPI) [13] includes a total of 120 parameters aimed at improving the standard of cancer care. In Europe, there are similar examples of widely implemented accreditation systems, such as the European Society of Medical Oncology (ESMO), which provides accreditation as part of a Designated Centre of Integrated Oncology and Palliative Care program that includes, among its 13 criteria, only one specific reference to the accreditation of personnel (physicians and nurses) in pain management [5]. The accreditation procedure is complemented by a set of quality indicators that are used as accreditation standards. These indicators are defined as measures that can facilitate the monitoring, evaluation, and improvement of the quality of care and organizational functions that affect patient outcomes. The indicators have been used for evaluating and reducing the gaps that exist in different healthcare practices [14]. Moreover, the indicators have been used systematically in oncology services and centers for several years. However, although it is true that there are guidelines and recommendations for pain management, there is a lack of specific accreditations that pertain to the management of chronic oncological pain.

The purpose of this study was to establish standards for the accreditation of oncological pain management in hospitals and cancer care units.

## 2. Materials and Methods

A mixed-methods study, including literature review, qualitative consensus analysis (Nominal Group and Delphi techniques) and a quantitative pilot study for validation of the standard was carried out between January and September 2019 to develop standards for accredited cancer pain management in healthcare organizations. This study included a review of the literature and qualitative techniques (Nominal Group and Delphi techniques) that were applied to identify consensus among all participants with regard to the process of addressing cancer pain. Finally, the validity and feasibility of the established standards and the accreditation process that were determined through self-assessment and external evaluation were assessed through a field study (Figure 1). This study was conducted by a CORE panel of experts (the study‘s faculty leadership) that comprised three anesthesiologists, three oncologists, two family physicians, one nurse, one psychologist, one patient representative, and four experts in qualitative research methods. The CORE panel of experts determined a set of criteria from the literature review and approved the final proposal of standards after applying the Delphi technique.

### 2.1. Literature Review (First Phase)

The literature search was carried out in the following databases: PubMed, Scopus, and Cochrane using the following Medical Subject Heading (MeSH) descriptors: “cancer”, “pain”, “quality of Health care”, “indicators”, and “therapy”. The search included only articles that were published in peer-reviewed journals, which accredited, developed, or validated the quality of criteria that were related to the clinical approach to the management of cancer pain. We included articles on the therapeutic management of cancer pain. In addition, a manual search was conducted on the websites of the Spanish Society of Palliative Care (SECPAL), Spanish Society of Medical Oncology (SEOM), Spanish Society of Pain (SED), Spanish Society of Radiation Oncology (SEOR), Cris against cancer, European Society of Medical Oncology (ESMO), Latin-American Guidelines for Cancer Pain Management, American Society Clinical Oncology (ASCO), Quality Oncology Practice Initiative (QOPI), Registered Nurses’ Association of Ontario (RNAO), National Institute for Health and CARE (NICE), and Scottish Intercollegiate Guidelines Network (SIGN) to identify literature on the management of cancer pain [15].

### 2.2. Nominal Group Technique (Second Phase)

The Nominal Group technique consists of a group meeting where experts participate in two phases: at the beginning to express their individual opinions and at the end to decide which ideas are considered most relevant (open debate and working group). The main goal of this qualitative technique is building a conceptual map of ideas [16,17,18]. To apply this technique, a session was held on 18 February 2019, where members of CORE panel experts discussed and, by consensus, the areas under which each of the quality criteria would be structured were established. It was based on the literature review and the ideas contributed during the Nominal Group by the CORE panel experts. During this work session, the moderator raised two key questions that helped to review the standards that need to be incorporated into an accreditation process for cancer pain management: How to identify services that adequately address chronic cancer pain? and what should be done that is not usually done? Subsequently, standards were developed for eight areas that were identified, and these underwent a second phase of evaluation by the CORE panel experts, thereby facilitating the incorporation of additional standards and evidence to support the proposed criteria.

### 2.3. Delphi Technique (Third Phase)

The result of this first consensus by the CORE panel experts (Questionnaire 0) (Appendix A
Appendix A) was evaluated by a group (*N* = 64) of professionals and patients who were drawn from the entire Spanish health system by using the Delphi technique in an online application (website designed for this purpose). Each of the participants received a personal invitation that allowed access to the website for voting in each of the two rounds. Professionals were addressed by their peers in the CORE panel experts, and patients and caregivers were contacted through the Spanish Patient Forum (a nonprofit, nongovernmental organization that provides counseling to patients). All participants had previously provided written informed consent that was obtained by using an online Web tool that granted the participants access to the app only after they had signed the informed consent agreement. There was an interval of approximately 1 month between each of the two rounds for obtaining consensus in the accreditation criteria. Through this technique, it was possible to evaluate (on a scale from 0 to 10 points; not important to most important) the criteria submitted by each of the participants in relation to their importance within an accreditation process in cancer pain as well as to ascertain their opinions and suggestions about this proposal (steps designed for the accreditation process). The implementation of the Delphi technique was supervised and approved by the Responsible Research Office of the Miguel Hernandez University (reference AUT.DPS.JMS.01.20).

The Delphi technique allows the votes of each participant to contribute to the establishment of consensus according to three levels: high agreement (which is included for the final assessment), low consensus agreement (which will be discarded), and doubtful consensus agreement, which will be subjected to a second round of voting to clarify whether they would be included. The cutoff point was determined by the degree of consensus reached in each round of voting. For the first round, a high or low consensus was considered to exist if the standard obtained an average score of more than 9.00 points or less than 8.00 points, respectively. For the second round of voting in the Delphi technique, only those standards that reached the cutoff point were selected for inclusion. These standards were further classified as structural, process, or outcome quality standards and prioritized as essential or recommended by the CORE panel experts.

### 2.4. Pilot Study (Fourth Phase)

The validity and feasibility of the proposed standards were evaluated in a pilot study that applied a self-evaluation procedure in eight hospitals from different Spanish cities between January and July 2020. The hospitals that participated in the piloting and validation process of the accreditation standard have an experience of more than 30 years in the treatment of oncological processes. They are national reference hospitals in different Spanish provinces. The self-evaluation was carried out using an online platform (http://acdon.es) that was designed specifically for this phase of the study wherein instructions for conducting the self-evaluation were included and information related to each standard were registered. The self-evaluation was carried out by staff from each hospital an in compliance with the patient data privacy regulations that were specified in accordance with the Spanish Biomedical Research Law (14/2017). None of the patient data was coded.

The standards related to the accreditation structure were evaluated, and were indicated by the presence or absence of information or documentation that proved compliance with the established standards. The standards related to the process were self-evaluated by auditing a random selection of medical records. The selection of clinical records included patients diagnosed with a tumor in any unit of the oncology department during the last 6 months. The review and evaluation of clinical records were based on a sampling of lot acceptance, wherein the thresholds of acceptance/rejection were established on the basis of predetermined levels of compliance. A sample size of 35 medical records, at a threshold of 80%, standard of 90%, for a β error of 0.2 (80% statistical power), and at a confidence level of 95% were determined. Thus, when the number of non-compliances, for each standard that required the audit of clinical records, was greater than 6, the whole lot was rejected.

After this task was completed, the individuals responsible for the self-evaluation of each hospital assessed the usefulness of the self-evaluation platform and the feasibility of the process, including the accessibility of reliable data in conformance with the proposed standards, through the Delphi technique.

## 3. Results

### 3.1. Literature Review (First Phase)

After screening a total of 563 records and assessing eligibility twenty-five articles, eight clinical practice guidelines, five organizational reports, and two quality standards reports related to the management of cancer pain were selected, a total of 28 records were excluded because did not addressed cancer related pain and were clinical trials.

### 3.2. Nominal Group Technique (Second Phase)

A first list of 37 criteria was obtained from the review of the scientific literature. This first list was submitted for discussion by the CORE panel expert using the Nominal Group technique. In this way, new criteria were added to the initial ideas.

Subsequently, at the first expert discussion meeting of the CORE panel experts, it was decided that the new quality criteria should be incorporated into the Delphi process.

### 3.3. Delphi Technique (Third Phase)

This resulted in the creation of Questionnaire 0 with 60 quality criteria which were distributed across eight areas: assessment and treatment (14 indicators), pharmacological treatment (8), non-pharmacological treatment (6), palliative care (6), coordination (8), teaching (7), patient safety (7), and patient satisfaction (3).

In the Delphi technique (Figure 2), 56 professionals and 8 patients responded in the first round, and 54 professionals and 8 patients in the second round, with a participation rate of 90% and 87%, respectively. In the first round, 23 criteria obtained a high consensus whereas 6 had a very low consensus, and therefore were not included for the second round of voting. The remaining standards were reevaluated due to doubtful consensus. As a result of Wave 1 of voting, participants provided 22 new suggestions that were reviewed by the CORE panel experts to decide if they should be included for assessment in the successive round of evaluation.

A third draft with 31 standards (of doubtful consensus agreement) that were extracted during the first Delphi round and the inclusion of 13 suggestions which were reviewed by the CORE panel experts’ group were included in the second round of voting. Finally, as a result of Round 2, 20 additional criteria that had a high consensus were included along with the initial 23 criteria of Round 1. This final proposal of 43 criteria was once again evaluated by the CORE panel experts, who considered the merger of two of these criteria because of their similarity and the exclusion of three criteria for their low accessibility and difficulty in measurement by using the available data. Finally, there was consensus on 39 standards. There were no differences in consensus between professionals and patients, except for standard 3.5, “The patient should evaluate with the professional which non-pharmacological alternatives can be added to the pharmacological treatment”, which was highly voted by caregivers and patients. Of these, 10 quality standards were considered to be essential by the CORE panel experts for the execution of future quality accreditation in the management of oncological pain. This implies that if the standard was not met, the accreditation was not possible.

The resulting ACDON accreditation system includes 18 outcome standards, 17 process standards, and 4 structure standards in the following areas of intervention. See Table 1.

### 3.4. Pilot Study (Fourth Phase)

In the self-evaluation process, 4 hospitals met all 10 standards that were considered essential. Only 2 met the 29 recommended standards. The overall degree of compliance with the essential standards was 90%, and compliance with the recommended standards was 78.45%. The average range of compliance was between 56.4 and 100% (Table 2). The area of intervention with least compliance were coordination and training, education and research (Table 3).

A total of 11 standards were 100% fulfilled in the 8 hospitals (37% essential and 63% recommended), and 24 standards, with over 80% (34% essential and 66% recommended). The standards for which the least compliance (Table 4) was noted were the implementation of a coordinated process to scheduled different services consultations the same day to reduce the discomfort associated with waiting lists, the implementation of actions aimed at preventing burnout among healthcare professionals, and the existence of a procedure for assessing patient experiences and the use of this information to improve the healthcare process. Appendix A
Appendix A presents the results of the self-assessment in each standard, classified by hospital, and as essential and recommended standards.

Based on these results, the CORE panel experts agreed to modify two standards to improve their comprehension: modified standards related to therapeutic guidelines for treatment and prevention of adverse events.

## 4. Discussion

Patient outcomes have become an integral endpoint for medical training and research. As healthcare systems transform into patient-centered, value-based accreditation and reimbursement units, the components of patient-centered outcomes become increasingly broad. The value criteria within patient-centered healthcare include outcomes that improve the quality of life after survival, relieve suffering, avoid morbidity, improve function, and restore personal dignity. To our knowledge, this is the first set of standards that were specifically designed for cancer pain management.

Over the past decade, multiple randomized controlled trials have demonstrated that timely involvement of specialists in palliative care, concurrent with oncological care, can improve health outcomes, including quality of life, mood, quality of end-of-life care, illness understanding, patient and caregiver satisfaction, and cost of care [19,20,21]. There are other accreditations for oncology services or centers such as ESMO, which through its 13 criteria seeks to improve the level of integration between palliative care and oncology services; however, they do not focus on pain management. On the other hand, the standard set of criteria proposed by this working group has 39 different recommendations for the adequate treatment of pain. This holistic focus includes different healthcare professionals involved in the patient journey; and has the added value to take into consideration perspectives of patients and caregivers in order to accomplish patient centered-care. One of the hospitals involved in this study, is a ESMO Designated Centre of Integrated Oncology and Palliative Care, making both accreditations compatible.

One of the ESMO criteria refers to the need for the center to have physicians and nurses who are experts in the evaluation and management of pain and other symptoms [22]. In a recent article, the authors measured this criterion only by the presence of an interdisciplinary team for palliative care and found that up to 90% of ESMO designated centers met this criterion. In the same research, another of its indicators refers to the fact that up to 30% of outpatients had not received an adequate assessment of their pain management before death [5]. This may suggest that the presence of an interdisciplinary team alone may not be sufficient for adequate control of pain. In another study, both inpatient and outpatient settings showed compliance with the National Cancer Comprehensiveness Network Clinical Practice Guidelines in Oncology for Adult Cancer Pain less than 70%, whereas more than one-third of patients continue to receive inadequate doses of analgesics [23].

With regard to the process and tools for accreditation, the participants pointed out the fact that, after the self-evaluation process, quality criteria in the care of oncological patients and the dynamics of teamwork were adequately visualized, including the variability in the approach to oncological pain, thereby facilitating the determination of the strengths and weaknesses and establishing improvement plans in a range of 15–20 min that were necessary for the assessment of the degree of compliance with each standard.

The accreditation standards presented in this article seek to establish quality criteria for a comprehensive approach to cancer pain that considers the professionals and the expectations of patients and their caregivers. The current focus of personalized medicine and patient-centered outcomes further individualizes the delivery and goals of medical care, which reinforces the existing need for adequate control of cancer pain [24]. The current approach seeks to bring together the expectations of patients and caregivers with those of professionals that may not be the same [25]. Therefore, we have included criteria that have reached a high level of consensus among all the involved parties, and in which the use of non-pharmacological treatment for the management and control of oncological pain should be highlighted. These therapies, which could be considered as an alternative care option despite their limited evidence, are well accepted by patients and their caregivers, such as acupuncture [26] and psychological support for family members [27].

The two main limitations of the study are. first, the proposed standards have only been applied in eight hospitals in Spain; therefore, much research is needed to establish whether the accreditation model is optimal for other health systems. Second, to reduce possible bias, a wide panel of different specialists and professionals for a multidisciplinary approach from different institutions of the Spanish national health system and patient associations were involved, but this accreditation model is tailored for the Spanish context.

## 5. Conclusions

Future research using the ACDON project could evaluate these standards of quality criteria for the care of patients with chronic oncological pain in various care centers of different healthcare systems. This will provide baseline information on the strengths and weaknesses of the involved services as well as the possibility of implementing the criteria and the frequency of their evaluation. This will be complemented with a work plan to improve the possible unaccomplished criteria in order to perform an appropriate translation of the criteria to clinical practice and to improve patient outcomes.

## Figures and Tables

**Figure 1 jpm-11-00102-f001:**
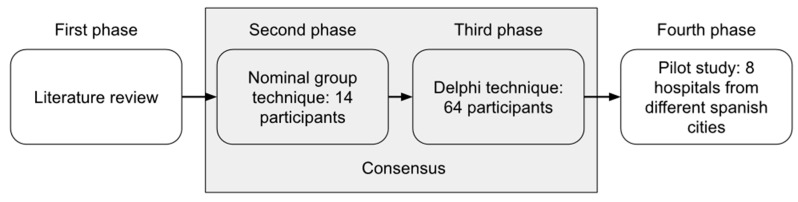
Phases of the study.

**Figure 2 jpm-11-00102-f002:**
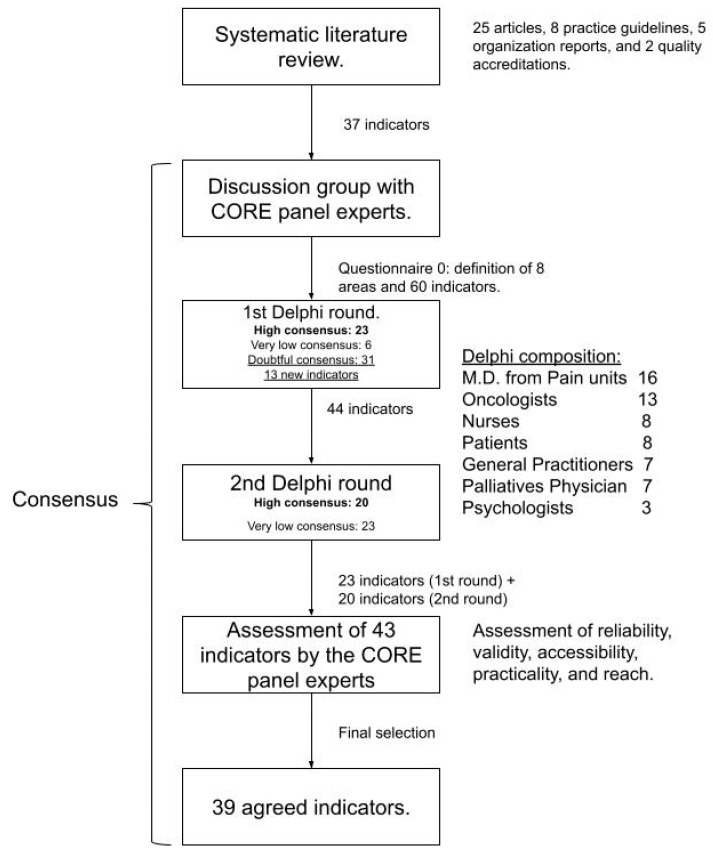
Flowchart of the Delphi process.

**Table 1 jpm-11-00102-t001:** ACDON accreditation system.

Area	Description	Number of Indicators
**Evaluation and assessment of oncological pain**	Quality criteria were incorporated with a focus on improving pain assessment using multidimensional scales and patient participation. In addition, assessment of family members and communication between the care team, caregivers, family members, and patients for adequate therapeutic adherence and improvement of clinical practice were included.	5
**Pharmacological treatment**	Treatment should be multimodal, through combining individualized strategies, treatment of underlying cause, modification of the underlying treatment, and the use of rescue treatment therapies and interventional techniques were included. Quality criteria to ensure pharmacological treatment has a continuous pattern that has to be agreed between professional and patient, including additional therapeutic demand (rescue) in the event of breakthrough pain were included.	6
**Non-pharmacological treatment**	Patients with chronic oncological pain requires a multidimensional approach. Quality criteria have been included to optimize pain management by considering proposals pertaining non-pharmacological measures that were aimed at improving the outcomes of cancer pain management with the ultimate goal of alleviating patient suffering.	5
**Palliative care**	Access to palliative care services for patients with chronic cancer pain is fundamental. Palliative care should be included in the routine management of patients to ensure that there is a continuous assessment, and that the patient can be duly informed in a timely manner.	4
**Coordination**	Effective coordination was highlighted for the optimal functioning of multidisciplinary teams. This coordination should be accompanied by information about the pain unit and other services that are available to the patient.	4
**Training, teaching, and research**	It would be necessary for the service to include the training for professionals to achieve adequate communication of bad news to patients and families as well as for activities to prevent burnout among physicians. Research has been highlighted as a necessary activity for improving care.	5
**Patient safety**	Quality criteria related to safe medication use, the prevention and control of drug interactions and adverse events as well as their timely reporting were included to ensure patient safety.	6
**Patient satisfaction**	All clinical efforts should be aimed at improving the patient’s quality of life and experience with health services. Results related to analgesia should be obtained from patients and relatives as well as the degree of dependence of the patient and the care and multimodal attention that was received.	4

**Table 2 jpm-11-00102-t002:** Results of the self-evaluation process.

Hospital	Essential Standards (Fulfilled/Total)	Recommended Standards (Fulfilled/Total)	Overall Compliance
Hospital 1	8/10	20/29	71.8%
Hospital 2	10/10	29/29	100%
Hospital 3	7/10	15/29	56.4%
Hospital 4	9/10	27/29	92.3%
Hospital 5	10/10	22/29	82.1%
Hospital 6	8/10	19/29	69.2%
Hospital 7	10/10	21/29	79.5%
Hospital 8	10/10	29/29	100%

**Table 3 jpm-11-00102-t003:** Compliance of standards according to areas.

Areas	Number of Standards	Overall Compliance
Assessment and counselling for cancer pain	5	85%
Pharmacological treatment	6	92.5%
Non-pharmacological treatment	5	80%
Palliative Care	4	94.7%
Coordination	4	65.6%
Training, education, and research	5	62.5%
Patient Safety	6	87.5%
Patient Satisfaction	4	78.1%

**Table 4 jpm-11-00102-t004:** Standards with least compliance.

Areas	Standards	Overall Compliance
Pharmacological treatment	A specific record should be kept in the digital medical record of the “itinerary” through the WHO analgesic ladder, as well as the reasons for it (at the discretion of your responsible physician), to facilitate the best coordination between the responsible team and the inter-current (emergency) teams.	62.5%
Non-pharmacological treatment	In case of pain that does not subside with non-invasive treatment, a non-pharmacological interventional technique should be proposed to the patient.	62.5%
Coordination	Patient information should be provided about the Pain Unit, its portfolio of services, accessibility and the strategy designed for the care of patients with chronic oncological pain in a proactive manner.	62.5%
	High resolution consultations should be implemented to reduce the discomfort associated with in-between-consultations.	50%
	A telephone follow-up plan should be established by calling the home of patients with cancer pain.	62.5%
Training, education, and research	Actions should be taken to prevent burnout among professionals. For example: Increased psychological support to reduce burnout among professionals.	25%
	The Pain Unit should participate in the development of new treatments and advances in the improvement of therapeutic effectiveness in the management of oncological pain, participating in research projects and clinical trials with direct translation to clinical practice.	50%
Patient Safety	The reasons for deciding not to follow the WHO pain management ladder should be recorded in the digital medical record.	62.5%
Patient Satisfaction	There must be a procedure for assessing the patient’s experience and using this information to improve the care process.	50%

## Data Availability

The data presented in this study are available on request from the corresponding author. The data are not publicly available due to privacy restrictions and protection of personal data.

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
