# Peer review of "Pain Standards for Accredited Healthcare Organizations (ACDON Project): A Mixed Methods Study"

_jpm, 2021, doi:10.3390/jpm11020102_

Round 1

Reviewer 1 Report

Pain in cancer care is a significantly undertreated concern and evidence-based standards to improve its management are warranted.  This study represents a very large undertaking by the research team across numerous stages from initial literature review to assessment of current implementation, which is to be commended.  However, more details and some clarifications about the processes will strengthen this article and its findings.  Recommendations on areas where more information or changes in the presentation of the methods and results are listed below:

Methods

  1. Section 2.2 Nominal group technique- more description of what the nominal group technique is would be useful or at least a reference for the first sentence of this section as this presents it as a supported technique
  2. Questionnaire 0 is presented in multiple different ways (e.g. Questionnaire 0 vs Questionnaire “0”)- suggest standardizing naming-
  3. Questions asked in the Delphi rounds are not described. What were the respondents asked to do?  Were they asked to support or not each criteria?  How was it scored (e.g. likert scale, yes/no etc) Were there open ended text boxes for additional criteria?  More information is needed
  4. Pg 4- end of section 2.3 ("For the second round of voting in the Delphi technique, only those standards that reached the cutoff point were selected for inclusion. These standards were further classified as structural, process, or outcome quality standards and prioritized as essential or recommended")
    - Were these standards classified with as structural, process or outcome as part as well as essential or recommended as part of the delphi or as a separate process?  Needs clarification
  5. Section 2.4- more information about 8 the hospitals were used in the study is needed. Why and how they were chosen?   Are they all based in cities?  Do they offer similar treatment options?  Etc
  6. Section 2.4- How were the medical records chosen? Randomly selected or sequential, etc?  Did they represent a range of cancer types, stages etc?

Results

  1. The organisation of this results section is a bit difficult to follow- I dont think the headings help the reader to follow the process that you've undertaken particularly as the headings dont correspond with anything from the methods
  2. Pg 4 line 187- “A total of 37 possible criteria were identified by means of the nominal group…”- I don’t really understand this- did these come from the literature review originally and then were discussed by nominal group? More details on how the criteria were determined from the literature review are needed.  What processes were undertaken to identify these as  possible criteria?  For so many studies, standards etc to have been included in the literature review, there is really no information on what processes were followed to determine the criteria that were the basis of the rest of the process.
  3. I found the distinction between the nominal group and the CORE group a bit hard to follow- are they different? eg in the flowchart there are 60 criteria carried forward from the CORE group but in the text it has the sentence …”total of 37 possible criteria were identified by means of the nominal group technique and included in the first draft for review and development by the core group.”. This may be easily clarified with a more clear description of the nominal group technique as mentioned in the method section comments.
  4. Flowchart 2- the number of indicators included in the flowchart at each stage does not seem to align with the text, which makes it hard to follow what was agreed upon and excluded at each stage. The small numbers next to the arrow between the boxes seem to refer to different things for different stages- or I am just not following the number of criteria that were considered and/or accepted at each stage of the consensus process
  5. Section 3.1-3.8- I wonder if these could be presented differently- the section numbers at all other parts of the manuscript refer to stages of the study and then suddenly the numbering refers to the findings/results.  I wonder if this could be better presented in a table to more easily show what is included in each criteria/section from a content perspective as well as how many outcome, process and structure standards there are for each content area
  6. Section 3.1 “ In the participants' opinion, one of the main causes of …” When was the participant’s opinion determined? Was this part of the delphi process?
  7. Table 3- Co-ordination row- what is a “high intensity consultation”?
  8. Training, education, and research row- “Actions should be taken to prevent burnout among professions”- is quite vague compared to the other criteria. I wonder if that’s why it scored lowest- possible discussion point?
  9. End of pg 8-pg 9- this paragraph is a very long sentence that is very hard to follow. I’m not sure what the result be presented in it was or how it was determined.  Is it more appropriate as a discussion point? 

Author Response

Dear Reviewer,

We appreciate very much the work done by the reviewers, particularly in this period we assume that is an additional effort as we are well aware of the difficulties arising from the COVID-19 outbreak. Many thanks. In addition, we appreciate all the suggestions and comments. We are sure they improve this work. In the manuscript, we have introduced track changes to make it easier to identify. In this letter, we comment one by one on these suggestions. Because of all the changes made a new table in the manuscript and a new table of supplementary material have been added. New references have also been added, which has changed the numbering.

Pain in cancer care is a significantly undertreated concern and evidence-based standards to improve its management are warranted.  This study represents a very large undertaking by the research team across numerous stages from initial literature review to assessment of current implementation, which is to be commended.  However, more details and some clarifications about the processes will strengthen this article and its findings.  Recommendations on areas where more information or changes in the presentation of the methods and results are listed below:

Thank you very much for these comments. We appreciate your interest in the study and we describe all the changes that have been made as a result of your suggestions.

Methods

1.Section 2.2 Nominal group technique- more description of what the nominal group technique is would be useful or at least a reference for the first sentence of this section as this presents it as a supported technique.

We have incorporated 3 references about the Nominal Group. We have tried to describe better what the purpose of this technique is.

2.Questionnaire 0 is presented in multiple different ways (e.g. Questionnaire 0 vs Questionnaire “0”)- suggest standardizing naming-

Thank you. We have always written in the text Questionnaire 0

3.Questions asked in the Delphi rounds are not described. What were the respondents asked to do?  Were they asked to support or not each criteria?  How was it scored (e.g. likert scale, yes/no etc) Were there open ended text boxes for additional criteria?  More information is needed

We have included a table of Supplementary Material with Questionnaire 0 of the Delphi technique. The supplementary material contains the instructions that participants received for answering each of the criteria.

4.Pg 4- end of section 2.3 ("For the second round of voting in the Delphi technique, only those standards that reached the cutoff point were selected for inclusion. These standards were further classified as structural, process, or outcome quality standards and prioritized as essential or recommended")
- Were these standards classified with as structural, process or outcome as part as well as essential or recommended as part of the delphi or as a separate process?  Needs clarification

The CORE panel expert has been in charge of reviewing all the results in each of the phases. Once the second round was finished, all the criteria regarding structure, process and result were reviewed and defined as essential or recommended. We have incorporated the role of the CORE panel expert as the author of these actions.

5.Section 2.4- more information about 8 the hospitals were used in the study is needed. Why and how they were chosen?   Are they all based in cities?  Do they offer similar treatment options?  Etc

The hospitals that participated in the pilot study of the accreditation standard have an experience of more than 30 years in the treatment of oncological processes. We have described these characteristics in the manuscript.

6.Section 2.4- How were the medical records chosen? Randomly selected or sequential, etc?  Did they represent a range of cancer types, stages etc?

The selection criteria referred to any patient diagnosed with a tumour in any unit of the oncology service during the last 6 months. We have described these characteristics in the manuscript.

Results

1.The organisation of this results section is a bit difficult to follow- I dont think the headings help the reader to follow the process that you've undertaken particularly as the headings dont correspond with anything from the methods

All sections have been standardised according to the phases of the study as shown in Figure 1.

2.Pg 4 line 187- “A total of 37 possible criteria were identified by means of the nominal group…”- I don’t really understand this- did these come from the literature review originally and then were discussed by nominal group? More details on how the criteria were determined from the literature review are needed.  What processes were undertaken to identify these as possible criteria?  For so many studies, standards etc to have been included in the literature review, there is really no information on what processes were followed to determine the criteria that were the basis of the rest of the process.

Based on a literature review, a first list of 37 criteria was obtained. This list was presented in the first qualitative work session of the CORE panel expert. The experts discussed the criteria and incorporated new ones, resulting in a set of 60 possible criteria that would be submitted to a Delphi consensus. These results have been detailed in the first and second phase of results.

I found the distinction between the nominal group and the CORE group a bit hard to follow- are they different? eg in the flowchart there are 60 criteria carried forward from the CORE group but in the text it has the sentence …”total of 37 possible criteria were identified by means of the nominal group technique and included in the first draft for review and development by the core group.”. This may be easily clarified with a more clear description of the nominal group technique as mentioned in the method section comments.

It´s true that this text is a bit confusing. Thanks for your comment. We have tried to clarify in section 3.2 the working group is the CORE panel expert whose composition and functions have already been explained. The Nominal group technique is the qualitative methodology used to discuss the inclusion or exclusion of possible criteria.

3.Flowchart 2- the number of indicators included in the flowchart at each stage does not seem to align with the text, which makes it hard to follow what was agreed upon and excluded at each stage. The small numbers next to the arrow between the boxes seem to refer to different things for different stages- or I am just not following the number of criteria that were considered and/or accepted at each stage of the consensus process

Sorry for this confusion. We have incorporated in flowchart 2 the whole process by which doubtful indicators from the first round move to the second round. Indicators with high consensus in the first round and indicators with high consensus in the second round were evaluated by the CORE panel expert for reliability, validity, accessibility and reach.

4.Section 3.1-3.8- I wonder if these could be presented differently- the section numbers at all other parts of the manuscript refer to stages of the study and then suddenly the numbering refers to the findings/results.  I wonder if this could be better presented in a table to more easily show what is included in each criteria/section from a content perspective as well as how many outcome, process and structure standards there are for each content area

We have made a summary table of the result of the ACDON accreditation system based on the reviewer's suggestion. This summary table is Table 1.

5.Section 3.1 “ In the participants' opinion, one of the main causes of …” When was the participant’s opinion determined? Was this part of the delphi process?

To making the summary table, we have removed this sentence which was confusing.

6.Table 3- Co-ordination row- what is a “high intensity consultation”?

Now is table 4. Sorry for the confusion. The correct term is “High Resolution consultation”. An ambulatory process of assistance fulfilled in a single day.

7.Training, education, and research row- “Actions should be taken to prevent burnout among professions”- is quite vague compared to the other criteria. I wonder if that’s why it scored lowest- possible discussion point?

We have incorporated an example of action to prevent burnout by offering more psychological support.

8.End of pg 8-pg 9- this paragraph is a very long sentence that is very hard to follow. I’m not sure what the result be presented in it was or how it was determined.  Is it more appropriate as a discussion point? 

We have incorporated this paragraph into the discussion section.

Reviewer 2 Report

Line

Comment

51

Please use the causes of pain as examples.  As written seems like all have to be present.

55

Remove “is suspected”.

57

“have not translated into clinical practice as expected”, not clear what is expected, maybe clarify “to produce adequate pain relief”.

67-68

Accreditations of what? Clinics, providers, standards? This introductory sentence needs to be more clear, although is not even needed.

89

Is it qualitative or mixed? A lit review is not qualitative.

186

What are the reasons to exclude papers?  Please indicate how many where excluded and why.

350

Remove “ some” and write:  “The two main limitations of the study are….”

Please review with a program to identify use of textual citations.

Author Response

Dear Reviewer,

We appreciate very much the work done by the reviewers, particularly in this period we assume that is an additional effort as we are well aware of the difficulties arising from the COVID-19 outbreak. Many thanks. In addition, we appreciate all the suggestions and comments. We are sure they improve this work. In the manuscript, we have introduced track changes to make it easier to identify. In this letter, we comment one by one on these suggestions. Because of all the changes made a new table in the manuscript and a new table of supplementary material have been added. New references have also been added, which has changed the numbering.

Line

Comment

51

Please use the causes of pain as examples.  As written seems like all have to be present.

We have clarified the sentence as follows:

The onset of chronic related cancer pain is associated with disease progression, stage of disease, and, most importantly, an impact on the patient's quality of life.[4,5]

55

Remove “is suspected”.

It´s removed

57

“have not translated into clinical practice as expected”, not clear what is expected, maybe clarify “to produce adequate pain relief”.

Thanks for the suggestion. We have used these terms

67-68

Accreditations of what? Clinics, providers, standards? This introductory sentence needs to be more clear, although is not even needed.

We have included the scope of the ASCO (American Society of Clinical Oncology) standard.

89

Is it qualitative or mixed? A lit review is not qualitative.

It is indeed a mixed study: literature review, qualitative consensus analysis and a quantitative pilot study for validation of the standard. We have incorporated it in the main text

186

What are the reasons to exclude papers?  Please indicate how many where excluded and why.

We have included the process of inclusion and exclusion of the documents included in the results of literature review process.

350

Remove “some” and write:  “The two main limitations of the study are….”

Thanks for the suggestion. We have used these terms

Please review with a program to identify use of textual citations.

We have incorporated the evaluation of the manuscript by an anti-plagiarism programme “Turnitin”. The percentage of similarity is 12% with other documents. This is due to similarities on the name of the scientific associations, quality rules and description of the scope of the quality accreditation procedure.